# Association of single nucleotide polymorphisms in the *NRF2* promoter with vascular stiffness with aging

**Sunao Shimizu**[1,2,3], **Junsei Mimura**[1], **Takanori Hasegawa**[4], **Eigo Shimizu**[5], **Seiya Imoto**[4,5], **Michiko Tsushima**[6], **Shuya Kasai**[1], **Hiromi Yamazaki**[1,2¤], **Yusuke Ushida**[2], **Hiroyuki Suganuma**[2], **Hirofumi Tomita**[6], **Masayuki Yamamoto**[7], **Shigeyuki Nakaji**[8], **Ken Itoh**[1,2] *

1 Department of Stress Response Science, Center for Advanced Medical Research, Hirosaki University Graduate School of Medicine, Hirosaki, Japan, 2 Department of Vegetable Life Science, Hirosaki University Graduate School of Medicine, Hirosaki, Japan, 3 Department of Nature & Wellness Research, Innovation Division, Kagome Co., Ltd. Nasushiobara, Tochigi, Japan, 4 Health Intelligence Center, The University of Tokyo, Minato-ku, Tokyo, Japan, 5 Human Genome Center, The Institute of Medical Science, The University of Tokyo, Minato-ku, Tokyo, Japan, 6 Department of Cardiology and Nephrology, Hirosaki University Graduate School of Medicine, Hirosaki, Japan, 7 Tohoku Medical Megabank Organization, Tohoku University, Sendai, Japan, 8 Department of Social Medicine, Hirosaki University Graduate School of Medicine, Hirosaki, Japan

¤ Current address: Department of Hematology-Oncology, Institute of Biomedical Research and Innovation, Foundation for Biomedical Research and Innovation at Kobe, Kobe, Japan
* itohk@hirosaki-u.ac.jp

**Data Availability Statement:** Data cannot be shared publicly because of the ethical concerns. Data are available from the Hirosaki University COI Program Institutional Data Access / Ethics

## Abstract

### Purpose

Pulse wave velocity (PWV), an indicator of vascular stiffness, increases with age and is increasingly recognized as an independent risk factor for cardiovascular disease (CVD). Although many mechanical and chemical factors underlie the stiffness of the elastic artery, genetic risk factors related to age-dependent increases in PWV in apparently healthy people are largely unknown. The transcription factor nuclear factor E2 (NF-E2)-related factor 2 (Nrf2), which is activated by unidirectional vascular pulsatile shear stress or oxidative stress, regulates vascular redox homeostasis. Previous reports have shown that a SNP in the *NRF2* gene regulatory region (−617C>A; hereafter called SNP−617) affects *NRF2* gene expression such that the minor A allele confers lower gene expression compared to the C allele, and it is associated with various diseases, including CVD. We aimed to investigate whether SNP−617 affects vascular stiffness with aging in apparently healthy people.

### Methods

Analyzing wide-ranging data obtained from a public health survey performed in Japan, we evaluated whether SNP−617 affected brachial-ankle PWV (baPWV) in never-smoking healthy subjects (n = 642). We also evaluated the effects of SNP−617 on other cardiovascular and blood test measurements.

Committee (contact via e-mail: coi@hirosaki-u.ac. jp) for researchers who meet the criteria for access to the data. Researchers need to be approved by research ethics review board at the organization of their affiliation.

**Funding:** This work was supported by "Center for Innovation Program" of the Japan Science and Technology Agency Grant Number JPMJCE1302 (https://projectdb.jst.go.jp/grant/JST-PROJECT-13423993/) and Joint Research Costs for the Department of Vegetable Life Science from Kagome Co., Ltd. The funders had no role in study design, data collection and analysis, decision to publish, or preparation of the manuscript. The Kagome Co., Ltd. provided support in the form of salaries for authors [SS, YU and HS] and the research costs as the Joint Research Costs, but did not have any additional roles in the study design, data collection and analysis, decision to publish, or preparation of the manuscript. The specific roles of these authors are articulated in the 'author contributions' section. SS: Sunao Shimizu, YU: Yusuke Ushida, HS: Hiroyuki Suganuma.

**Competing interests:** The authors have declared that no competing interests exist. This does not alter our adherence to PLOS ONE policies on sharing data and materials.

## Results

We have shown that not only AA carriers (n = 55) but also CA carriers (n = 247) show arterial stiffness compared to CC carriers (n = 340). Furthermore, SNP−617 also affected blood pressure indexes such as systolic blood pressure and mean arterial pressure but not the ankle brachial pressure index, an indicator of atherosclerosis. Multivariate analysis showed that SNP−617 accelerates the incremental ratio of baPWV with age.

## Conclusions

This study is the first to show that SNP−617 affects the age-dependent increase in vascular stiffness. Our results indicate that low NRF2 activity induces premature vascular aging and could be targeted for the prevention of cardiovascular diseases associated with aging.

## Introduction

Cardiovascular disease (CVD) is a major cause of death worldwide. The Global Burden of Disease, Injuries, and Risk Factors Study (GBD) reported that CVD killed 17.8 million people worldwide in 2017, corresponding to 31% of all deaths [1]. Furthermore, the number of deaths due to CVD increased by 21% from 2007 to 2017 [2]. Although epidemiological studies have identified various risk factors, such as lifestyle and genetic factors, for CVD, the most important risk factor is aging [3]. Alteration of blood vessels with aging, which manifests as vascular dilation, thickening of intima and media (IM), endothelial dysfunction or arterial stiffening, has been suggested to contribute to the increased risk of CVD due to aging [4, 5]. These factors could appear in the absence of overt clinical CVD during aging and are interrelated, for example, such that the age-associated increase in IM thickening is associated with an increase in vessel stiffness [6]. The obstruction of blood vessels is usually considered to be due to atherosclerosis characterized by lipid plaque accumulation that leads to tissue ischemia and plaque rupture. On the other hand, arteriosclerosis, which means arterial stiffness, is generally due to the degeneration of the media of vessels. Both conditions are the major risk factors for CVD and increase with age [4]. Pulse wave velocity (PWV), an index of the stiffness of the elastic artery, increases with age and is increasingly recognized as an independent risk factor for CVD associated with aging [7]. Increased PWV has been found in populations with little or no atherosclerosis, indicating that vascular stiffening can occur independently of atherosclerosis [7]. Although many mechanical and chemical factors are known to contribute to the stiffness of the conduit artery, the underlying factors related to the age-dependent increase in PWV, especially the genetic factors, are largely unknown [7].

Oxidative stress in vascular endothelial cells increases with age. Carotid arteries of aged rhesus macaques show significant oxidative stress as indicated by the increased 8-iso- prostaglandin F2α (PGF2α) and 4-hydroxy-2-nonenal (4-HNE) content and decreased glutathione and ascorbate levels compared to vessels of young macaques [8]. Similarly, the intracellular level of thiobarbituric acid reactive substance (TBARS), a lipid oxidation indicator, was increased in aged compared to young human vascular endothelial cells [9]. The transcription factor nuclear factor E2 (NF-E2)-related factor 2 (Nrf2) plays an important role in combating oxidative stress in metazoans of the animal kingdom. Normally, Nrf2 is kept in an inactivated state trapped in the E3 ligase adaptor Kelch-like ECH-associated protein 1 (Keap1) and is subsequently degraded in the cytoplasm by the proteasome. However, when reactive cysteine residues of

Keap1 protein are modified by oxidative stress or electrophilic substances, degradation of Nrf2 is inhibited, and undegraded Nrf2 translocates to the nucleus to bind to the antioxidant response element (ARE). Importantly, Nrf2 activity in the endothelium is regulated by laminar shear stress such that Nrf2 in the endothelium is high in the straight blood vessel but low in the curved or branching points of the blood vessels [10]. Nrf2 regulates the oxidative stress response in the body by inducing downstream antioxidant genes, such as those involved in glutathione (GSH) synthesis and heme oxygenase-1 (HO-1) [11–13]. Of note, Nrf2 activity decreases with age [14–17]. Nrf2 ameliorates endothelial dysfunction in diabetes and under condition of oxidative stress [18] and increases the amount of bioavailable nitric oxide (NO), which is a major vasodilator [19]. Oxidative stress is also known to induce vascular senescence [20], which is important for vascular aging. Nrf2 is inhibited in Hutchinson-Gilford progeria syndrome and premature senescence syndrome in humans [21] and has been proposed as a senescence modulator [22, 23]. Recently, Nrf2 was shown to be responsible for vascular senescence that occurs in the absence of CR6 interacting protein 1 (CRIP1) via the regulation of Sirt3 [24]. However, little evidence has shown that Nrf2 protects against vascular aging in humans.

Multiple single nucleotide polymorphisms (SNPs) have been detected in the human *NRF2* gene [25, 26]. The *NRF2* gene SNP (−617C>A; rs6721961), which is located in the promoter region, affects the transcriptional level of *NRF2* and thereby the protein level and downstream gene expression [27, 28]. It has been reported that SNP−617 is associated with various diseases, such as acute lung injury [27, 29], asthma [30], adenocarcinoma [31], breast cancer [32] and noise-induced hearing loss [33]. Furthermore, SNP−617 is associated with cerebrovascular disease and increasing blood pressure (BP) in a Finnish cohort [34] and hemodialysis patients in Japan [35], but the association between SNP−617 and arterial stiffness remained unclear. This study aimed to investigate the association of SNP−617 with arterial stiffness and BP in healthy Japanese people. This study for the first time showed that SNP−617 affects the age-dependent increase in vascular stiffness by analyzing well-controlled healthy never-smokers. This result offers important evidence that continuous activation of Nrf2 could improve arterial stiffness and exert a preventative effect on cardiovascular diseases associated with aging via improvement in arterial stiffness.

## Material and methods

### Study subjects

The subjects were participants in the Iwaki Health Promotion Project conducted by Hirosaki University from 2014 to 2017, which exclusively targeted the residents of the Iwaki area, Hirosaki city, Aomori prefecture, Japan [36, 37]. Out of the participants (n = 1816), 16 people did not consent to the genotyping, so 1800 were considered as the base subjects. Of the base subjects, 24 participants with missing questionnaires regarding clinical and smoking history and missing values for baPWV and 1129 people with a clinical history of malignant tumor, cardiovascular disease, renal or liver dysfunction and diabetes, medication history of hypertensive medicine or smoking history were excluded, leading to a total of 647 subjects. Of the 647 participants, those in their 80s were excluded because they did not include AA carriers, and the risk factors for CVD may become very complex beyond the age of 80 [38] (Fig 1). This research was conducted after being approved by the ethics committee of the Hirosaki University Graduate School of Medicine as a test design in accordance with the Declaration of Helsinki and recruiting subjects with written informed consent.

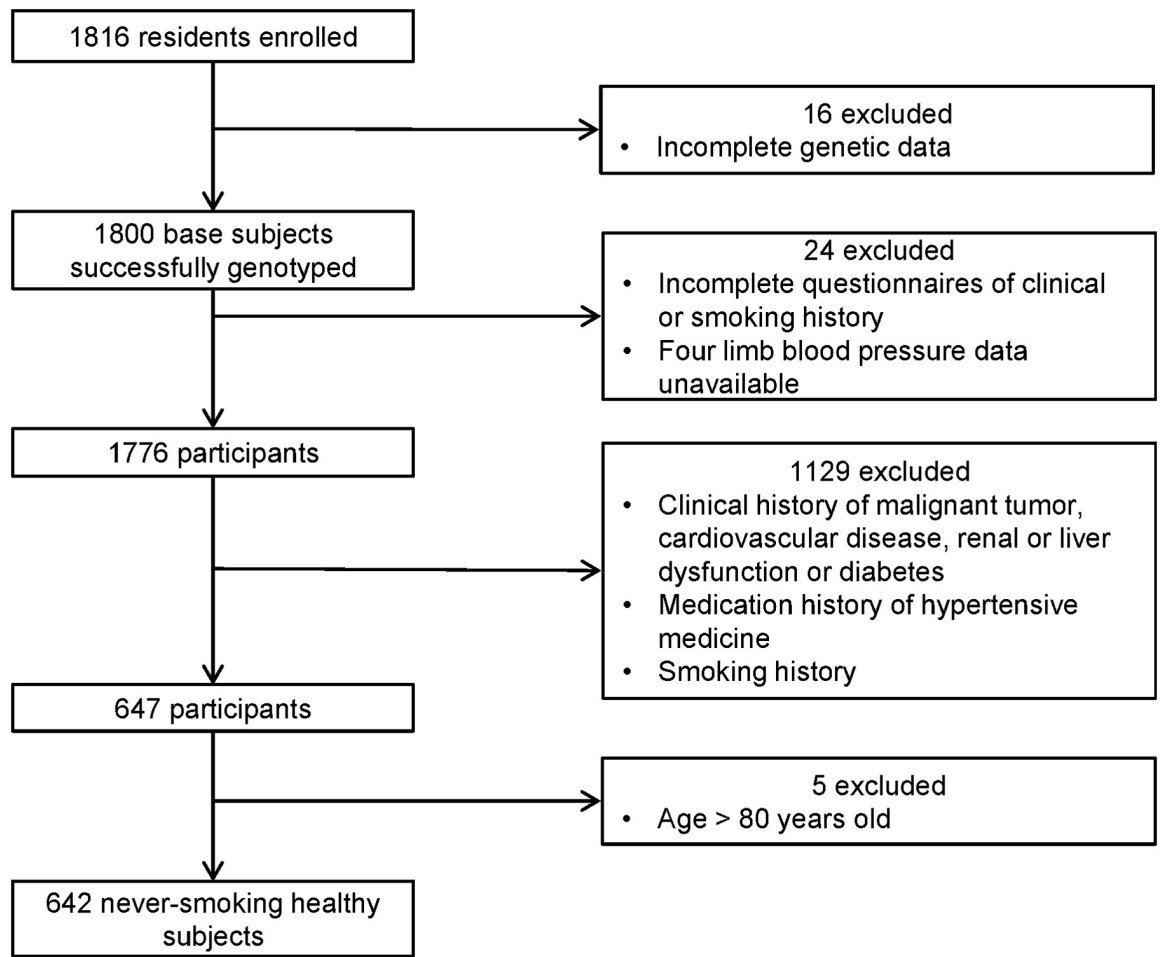

**Fig 1. Flow of the participants according to inclusion and exclusion criteria.** A total of 642 people were set as never-smoking subjects from all residents who participated in the Iwaki Health Promotion Project from 2014 to 2017.

### *NRF2* promoter SNP−617 (rs6721961) genotyping

Genotypes of SNP−617 were determined by whole genome sequencing, with imputation from Japonica Array (Toshiba, Tokyo, Japan) [39] consisting of population-specific SNP markers designed from the 1070 whole genome reference panel [40] and TaqMan PCR. Whole genome sequencing and imputation were performed by Takara Bio corporation (Shiga, Japan) and Toshiba corporation, respectively. For the Japonica Array, DNA was purified from peripheral whole blood using a QIAamp® 96 DNA Blood Kit (QIAGEN, Hilden, Germany). DNA was extracted from plasma pellets for whole genome sequencing. PCR genotyping of SNP−617 was performed as described in a previous report [41]. Briefly, an aliquot of genomic DNA samples was subjected to allele-specific PCR analysis by using the following PCR primers and TaqMan probes: rs6721961-F: 5'−CAG TGG GCC CTG CCT AG−3', rs6721961-R: 5'−TCA GGG TGA CTG CGA ACA C−3', rs6721961-T: 5'−[FAM]−TGT GGA CAG C<u>T</u>C CGG CAG− [MGBEQ]−3', rs6721961-G: 5'−[HEX]−TGG ACA GC<u>G</u> CCG GCA G−[MGBEQ]−3' (SNP sites are underlined). Real-time PCR was carried out by using SsoAdvanced Universal Probes Supermix (Bio-Rad) and a CFX 384 real-time PCR detection system (Bio-Rad) according to the manufacturer's protocol. For participants in 2014, whole genome data and

imputation data were used for genotyping, and PCR analysis was performed when there were mismatches between them. For participants between 2015 and 2017, PCR genotyping and imputation data were used for genotyping, and PCR analysis was used when there were mismatches between them.

## Measurement of four limb blood pressure

Brachial-ankle PWV (baPWV), systolic BP (SBP), and diastolic BP (DBP) were measured using an automatic waveform analyzer, BP-203RPE (Colin, Komaki, Japan). The ankle brachial pressure index (ABI) was calculated as the ratio of the ankle SBP to the brachial SBP. Pulse pressure (PP) was calculated as the difference between SBP and DBP. Mean arterial pressure (MAP) was calculated as the sum of one-third of SBP and two-thirds of DBP.

## Measurements of blood samples and questionnaires

The blood samples were collected between 6 AM and 9 AM after fasting for at least 9 h. Blood samples were obtained by venipuncture to measure erythrocyte, hemoglobin, hematocrit, total protein, aspartate transaminase (AST), alanine aminotransferase (ALT), gamma-glutamyl transpeptidase (©-GTP), blood urea nitrogen (BUN), creatinine (CRE), fasting glucose (Glc), glycated hemoglobin (HbA1c), insulin, C-peptide, triglyceride (TG), total cholesterol, high-density lipoprotein-cholesterol (HDL), low-density lipoprotein-cholesterol (LDL), iron, ferritin and total bilirubin. HOMA-IR was calculated by the following formula: HOMA = IR = insulin × Glc/405. The L/H ratio was calculated as the ratio of LDL to HDL. All measurements of blood samples were performed by LSI Medience Corporation. Data of self-reported sex, age, clinical history, medication history of hypertensive medicine and smoking history were determined from questionnaires (details of the questionnaires are in S1 File).

## Statistical analysis

The data are shown as the median/interquartile range (IQR). The Hardy-Weinberg equilibrium, and the comparison of allele frequencies between different databases, comparison of exclusion rate by sex and age and comparison of smoking rate by sex were analyzed by the $\chi^2$ test. The comparison of the mean values between the different genotype groups was analyzed by the Kruskal-Wallis (K.W.) test with post hoc analysis using the Holm method. Standardized multiple linear regression analysis was performed to determine the association between SNP −617, baPWV and age. Statistical significance was defined as a *p* value of <0.05. All statistical analyses were performed with R software version 3.6.2.

## Results

### Incidence of *NRF2* promoter SNP−617

Table 1 shows the prevalence of the *NRF2* promoter SNP (rs6721961, −617C>A: homozygote CC, heterozygote CA and homozygote AA; hereafter called SNP−617) and the calculated frequencies of the minor allele frequency (MAF) in this study and those from other databases. In this study, the frequencies of the genotypes were 51.5% for CC (n = 927), 40.2% for CA (n = 724), and 8.3% for AA (n = 149). The Hardy-Weinberg equilibrium test revealed that the MAF (0.284) in this study was in the equilibrium state (*p* = 0.950). The MAF value was slightly higher than that of jMorp (Japanese Multi Omics Reference Panel) 4.7KJPN (0.247) [42] and 1KGP (1000 Genomes Project) East Asian (0.243) but much higher than that of 1KGP Global (0.145) [43], with a significant difference (Iwaki Health Promotion vs jMorp 4.7KJPN, *p* <0.001; vs 1KGP East Asian, *p* = 0.004; vs 1KGP Global, *p* <0.001). The MAF values for SNP

**Table 1. The frequency of the *NRF2* SNP (−617C>A).**

| | MAF | Total | n (%) | | | P value[b] |
|---|---|---|---|---|---|---|
| | | | CC | CA | AA | |
| **Iwaki Health Promotion** | | | | | | |
| Real number | 0.284 | 1800 | 927 (51.5) | 724 (40.2) | 149 (8.3) | |
| Estimated number[a] | | | 923 (51.3) | 732 (40.7) | 145 (8.1) | 0.950 |
| **Data base** | | | | | | |
| jMorp 4.7KJPN[a] | 0.247 | 4773 | 2703 (56.6) | 1777 (37.2) | 292 (6.1) | <0.001 |
| 1KGP East Asian[a] | 0.243 | 1008 | 578 (57.3) | 371 (36.8) | 60 (5.9) | 0.004 |
| 1KGP Global[a] | 0.145 | 5008 | 3661 (73.1) | 1242 (24.8) | 105 (2.1) | <0.001 |

[a] The estimated frequency numbers of the Iwaki Health Promotion and the databases were calculated provided that each allele was in Hardy-Weinberg equilibrium.

[b] P values were calculated by $X^2$test.

Abbreviations: MAF, minor allele frequency; jMorp, Japanese Multi Omics Reference Panel; 1KGP, 1000 genomes project.

−617 in Japan are significantly higher than those for 1KGP Global, indicating a characteristic SNP−617 minor allele distribution in Japan. The SNP−617 allele frequencies between male and female subjects in this study were nearly identical (S1 Table).

## Association of *NRF2* promoter SNP−617 with arterial stiffness and blood pressure in never-smoking healthy subjects

To investigate whether SNP−617 is associated with the risk of CVD, we compared baPWV, which is an indicator of arterial stiffness, among CC, CA and AA carriers. The median baPWV was higher in the order of AA, CA and CC (median/IQR: CC, 1230/1092-1450 cm/s; CA, 1285/1106-1510 cm/s; AA, 1317/1169-1589 cm/s, respectively), and there was a significant difference between CC and AA carriers ($p < 0.05$) (Fig 2A). Next, we compared ABI, which indicates stenosis of the vasculature often due to atherosclerosis [44]. Interestingly, there were no significant differences between the genotypes in ABI values, indicating that the SNP−617 AA allele specifically affects arteriosclerosis but not atherosclerosis ($p = 0.84$) (Fig 2B). It is well known that baPWV associates with SBP [7]. To investigate the association of SNP−617 with SBP, DBP, MAP and PP, we compared these parameters among CC, CA and AA carriers. As expected, the median SBP, MAP and PP were higher in the order of AA, CA and CC, with significant differences between CC and CA in SBP, MAP and PP ($p = 0.006, 0.037$ and $0.042$, respectively), and between CC and AA in MAP ($p = 0.046$), but no significant difference in DBP was noted between the individual carriers (Fig 2C–2F). In addition, we also found some sex differences in these parameters (S2 Table). In female subjects, the K.W. test verified that there were significant differences in baPWV, SBP, MAP and PP ($p = 0.039, 0.028, 0.048$ and $0.038$, respectively), but there was a significant difference between the individual carriers in only baPWV ($p = 0.049$ for CC vs AA). On the other hand, there was no significant difference in any of the four-limb blood pressure measurements in male subjects (S2 Table).

## Association of *NRF2* promoter SNP−617 with general condition in never-smoking healthy subjects

Next, we investigated whether biomarkers that reflect the general condition of the subjects, including those associated with the risk of developing CVD, were affected by SNP−617. The association between SNP−617 and the biomarkers is shown in Table 2. As a result, the K.W. test verified that there were significant differences in creatine and HDL cholesterol ($p = 0.036$

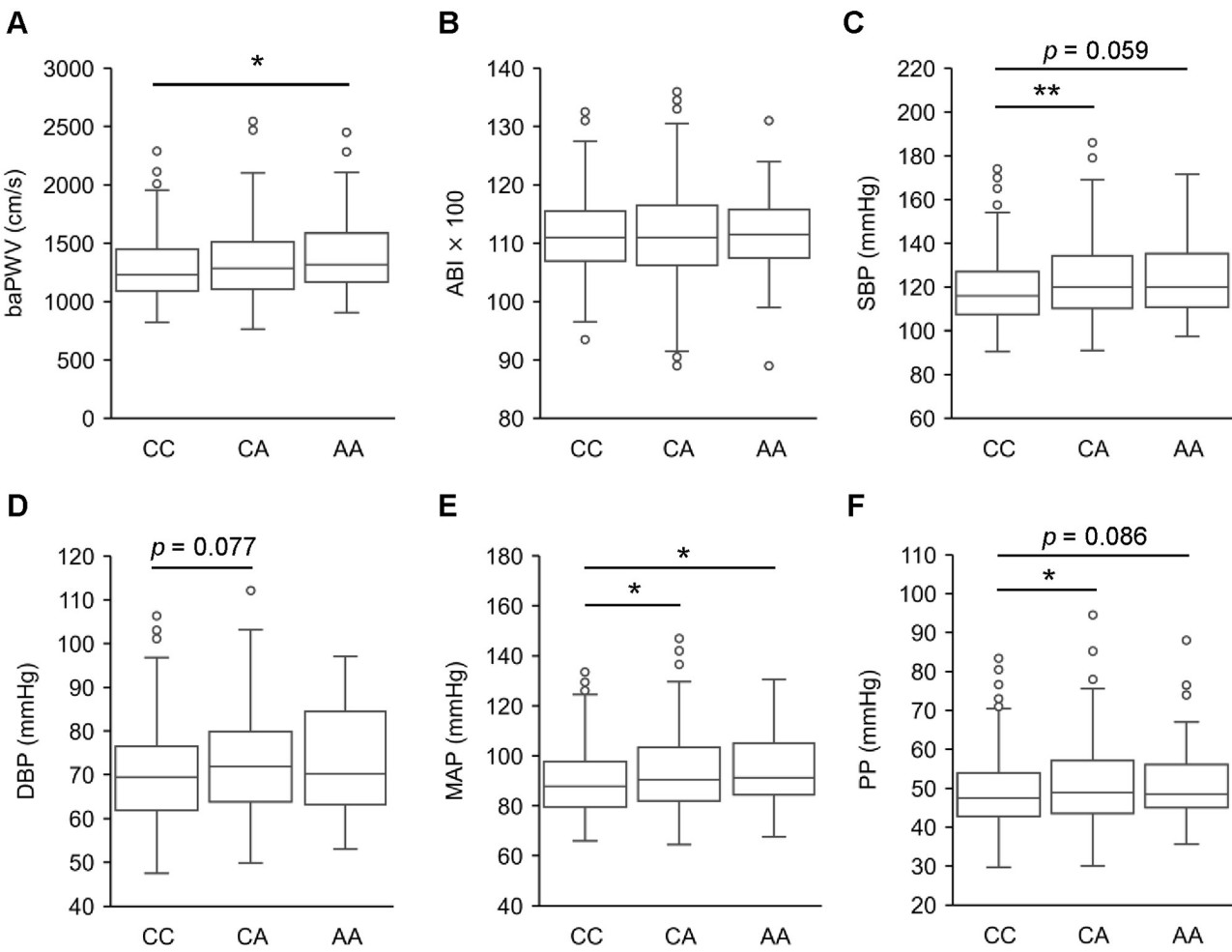

**Fig 2. Arterial stiffness and blood pressure were affected by *NRF2* SNP−617.** The boxplots with median, quartiles and min-max values of A) baPWV, B) ABI, C) SBP, D) DBP, E) MAP and F) PP were compared with respect to *NRF2* SNP−617. The data were analyzed by the Kruskal-Wallis test with a post hoc test by Holm. *$p < 0.05$, **$p < 0.01$.

and 0.024, respectively). The median creatine level in AA carriers was significantly lower than that in CC and CA carriers ($p = 0.045$ and 0.045, respectively), but there was no significant difference between the individual carriers in HDL cholesterol (Table 2). In addition, we investigated whether there was a difference between the results by sex. In female subjects, the K.W. test, as with no stratification, verified that there were significant differences in creatinine and HDL cholesterol ($p = 0.043$ and 0.029, respectively). The median creatine level in AA carriers was significantly lower than that in CC carriers ($p = 0.030$), and the median HDL cholesterol level in CA carriers was significantly lower than that in CC carriers ($p = 0.045$) (S3 Table). In male subjects, there was no significant difference in the biomarkers (S4 Table).

## Association of *NRF2* promoter SNP−617 with increasing arterial stiffness due to aging in never-smoking healthy subjects

To investigate whether SNP−617 affects the age-dependent increase in arterial stiffness, multivariate analysis of baPWV in relation to SNP−617 and age was performed. Table 3 shows standardized multiple linear regression analysis on baPWV and age, including SNP−617 and its

**Table 2. The median and IQR of general characteristics in never-smoking healthy subjects.**

| Characteristics | | median/IQR | | | P value[a] | | | |
|---|---|---|---|---|---|---|---|---|
| | | CC | CA | AA | K.W. | CC vs CA | CC vs AA | CA vs AA |
| Number of subjects | | 340 | 247 | 55 | | | | |
| Age | | 44/34-59 | 46/34-62 | 51/34-61 | 0.460 | - | - | - |
| Erythrocyte | (10,000/μl) | 446/422-473 | 448/423.5–480 | 438/413-459 | 0.202 | - | - | - |
| Hemoglobin | (g/dL) | 13.2/12.5–14.1 | 13.4/12.6–14.3 | 13.1/12.4–13.9 | 0.413 | - | - | - |
| Hematocrit | (%) | 42.1/40-44.6 | 42.4/40.4–44.8 | 42/39.8–44 | 0.334 | - | - | - |
| Total protein | (g/dL) | 7.3/7.0–7.6 | 7.3/7.0–7.5 | 7.4/7.1–7.7 | 0.250 | - | - | - |
| AST | (IU/L) | 19/16-23 | 20/16-24 | 19/17-23 | 0.913 | - | - | - |
| ALT | (IU/L) | 15/12-21 | 16/12-21 | 15/11-19 | 0.458 | - | - | - |
| γ-GTP | (IU/L) | 18/13-26 | 17/14-24 | 17/13-24 | 0.843 | - | - | - |
| BUN | (mg/dL) | 13.2/11.1–15.8 | 13.5/10.7–16.0 | 12.8/10.2–14.4 | 0.438 | - | - | - |
| Creatinine | (mg/dL) | 0.63/0.58–0.71 | 0.65/0.57–0.74 | 0.59/0.53–0.68 | 0.036 | 0.499 | 0.045 | 0.045 |
| Fasting glucose | (mg/dL) | 78/72-85 | 79/73-85 | 80/76-85.5 | 0.449 | - | - | - |
| HbA1c | (%) | 5.6/5.4–5.8 | 5.6/5.4–5.8 | 5.6/5.3–5.8 | 0.749 | - | - | - |
| Insulin | (μU/mL) | 4.0/3.0–5.2 | 4.1/3.0–5.5 | 3.9/2.9–5.5 | 0.785 | - | - | - |
| HOMA-IR | | 0.76/0.57–1.05 | 0.82/0.55–1.12 | 0.80/0.55–1.1 | 0.821 | - | - | - |
| C-peptide | (ng/mL) | 0.9/0.7–1.1 | 0.9/0.7–1.1 | 0.8/0.7–1.1 | 0.288 | - | - | - |
| Triglyceride | (mg/dL) | 65/47-93 | 66/48.5–95.5 | 68/54-93 | 0.686 | - | - | - |
| Total cholesterol | (mg/dL) | 195/171-220 | 197/171-215 | 205/174-223 | 0.395 | - | - | - |
| HDL cholesterol | (mg/dL) | 68/57-78 | 64/55-74 | 69/58-85 | 0.024 | 0.071 | 0.259 | 0.071 |
| LDL cholesterol | (mg/dL) | 110/91-131 | 110/91-130 | 112/91-132 | 0.946 | - | - | - |
| LH ratio | | 1.59/1.28–2.15 | 1.71/1.37–2.14 | 1.59/1.25–1.94 | 0.092 | 0.144 | 0.714 | 0.250 |
| Iron | (μg/dL) | 97/75-119 | 94/72-121 | 95/71-121 | 0.904 | - | - | - |
| Ferritin | (ng/mL) | 49.5/20.3–102 | 53.5/22.0–108 | 54.1/14.9–85.5 | 0.650 | - | - | - |
| Total bilirubin | (mg/dL) | 0.8/0.7–1.0 | 0.8/0.6–1.0 | 0.8/0.7–1.0 | 0.545 | - | - | - |

[a] *P* values were calculated by Kruskal-Wallis (K.W.) test with post-hoc test by Holm.

Abbreviations: IQR, interquartile range; AST, aspartate transaminase; ALT, alanine aminotransferase; γ-GTP, gamma-glutamyl transpeptidase; BUN, blood urea nitrogen; HbA1c, glycated hemoglobin; HOMA-IR, homeostasis model assessment of insulin resistance; HDL, high-density lipoprotein-cholesterol; LDL, low-density lipoprotein-cholesterol; L/H ratio, LDL/HDL ratio.

interaction. In all never-smoking healthy subjects, the baPWV of AA carriers was significantly higher and that of CA carriers was slightly higher than that of CC carriers (std. β ± SD/*p* value: CC vs CA, 0.11 ± 0.06/0.051; CC vs AA, 0.30 ± 0.10/0.003, respectively). Furthermore, AA carriers had a trend toward a higher increasing rate of baPWV with age than CC carriers (std. β ± SD/*p* value: 0.20 ± 0.10/0.052) (Fig 3A). It was reported that after 60 years of age, PWV rises sharply [45]. Therefore, we investigated whether there was a difference between the younger subjects (<60 years old) and the older subjects (≥60 years old) in the association of SNP−617 with an age-dependent increase in baPWV. For those aged under 60 years, the baPWV of the AA carriers was significantly higher and that of the CA carriers had a higher trend than that of the CC carriers (std. β ± SD/*p* value: CC vs CA, 0.14 ± 0.08/0.08; CC vs AA, 0.38 ± 0.14/0.007, respectively). Furthermore, CA and AA carriers had a significantly higher rate of age-dependent increase in baPWV than CC carriers (std. β ± SD/*p* value: 0.21 ± 0.08/0.01; 0.48 ± 0.13/<0.001, respectively) (Fig 3B). However, in those aged over 60, there was a significant difference in only age (std. β ± SD/*p* value: 0.35 ± 0.11/0.003) (Fig 3C).

**Table 3. Standardized multiple linear regression analysis of *NRF2* SNP−617, baPWV and age.**

| Age range | CC/CA/AA | Total | Characteristics | std. β [a] | SD | P [a] | r² |
|---|---|---|---|---|---|---|---|
| All age | | | | | | | |
| n | 340/247/55 | 642 | NRF2 SNP-617 CC vs CA | 0.114 | 0.058 | 0.051† | 0.523 |
| | | | NRF2 SNP-617 CC vs AA | 0.302 | 0.101 | 0.003** | |
| | | | Age | 0.663 | 0.039 | <0.001*** | |
| | | | Int. (SNP-617 CC vs CA: Age) | 0.072 | 0.058 | 0.211 | |
| | | | Int. (SNP-617 CC vs AA: Age) | 0.196 | 0.101 | 0.052† | |
| <60 | | | | | | | |
| n | 257/170/40 | 467 | NRF2 SNP-617 CC vs CA | 0.142 | 0.080 | 0.078† | 0.350 |
| | | | NRF2 SNP-617 CC vs AA | 0.379 | 0.139 | 0.007** | |
| | | | Age | 0.433 | 0.051 | <0.001*** | |
| | | | Int. (SNP-617 CC vs CA: Age) | 0.206 | 0.081 | 0.011* | |
| | | | Int. (SNP-617 CC vs AA: Age) | 0.481 | 0.130 | <0.001*** | |
| ≥60 | | | | | | | |
| n | 83/77/15 | 175 | NRF2 SNP-617 CC vs CA | 0.076 | 0.155 | 0.627 | 0.082 |
| | | | NRF2 SNP-617 CC vs AA | 0.396 | 0.279 | 0.157 | |
| | | | Age | 0.347 | 0.114 | 0.003** | |
| | | | Int. (SNP-617 CC vs CA: Age) | -0.172 | 0.156 | 0.271 | |
| | | | Int. (SNP-617 CC vs AA: Age) | -0.261 | 0.280 | 0.353 | |

[a] The $r^2$, std. β and $P$ values were calculated by the standardized multiple linear regression analyses on baPWV under 5 characteristics, i.e., NRF2 SNP−617 (CC vs CA), SNP−617 (CC vs AA), age, interaction between SNP−617 (CC vs CA) and age, and interaction between SNP−617 (CC vs AA) and age.

†$P$ <0.1,

*$P$ <0.05,

**$P$ <0.01,

***$P$ <0.001.

Abbreviations: Int., interaction; $r^2$, coefficient of determination; std. β, standardised partial regression coefficient; SD, standard deviation.

## Discussion

### Regional differences in *NRF2* promotor SNP−617

The MAF of SNP−617 A is 0.284, and SNP−617 AA carriers represent 8.3% of the IWAKI district of Hirosaki (Table 1), which is greater than the MAF of 0.247 in jMorp 4.7 KJPN and of 0.243 in 1KGP East Asian and the SNP−617 A/A homozygote ratios of 5.4% and 5.9% in a healthy normal population in previous publications [28, 41]. The reason for the higher occurrence of the SNP−617 A allele in this study is not clear at present. As the participants of the IWAKI health promotion project examined in this study cover only approximately 20% of all residents in the district, the results may be somewhat biased. It is known that recent natural selection contributes to the bias of certain SNPs within the Japanese population [46]. It is of note that the Hirosaki "TSUGARU" region covering the Iwaki district recently suffered from extensive famine (i.e., the five biggest famines) because of cold weather and a scarcity of food during the Edo period (i.e., approximately 1,600~1850 A.C.). For example, one-third of the residents died in the Tenmei famine, which is one of the largest famines [47]. It is an interesting hypothesis that the increase in SNP−617 MAF occurred by recent natural selection. One report from the Nagoya cohort showed a high MAF of 0.275 and AA carriers accounting for 10.1%. Although SNP−617 in their cohort is not in Hardy-Weinberg equilibrium, the SNP −617 genotype distribution might show regional differences in Japan [48].

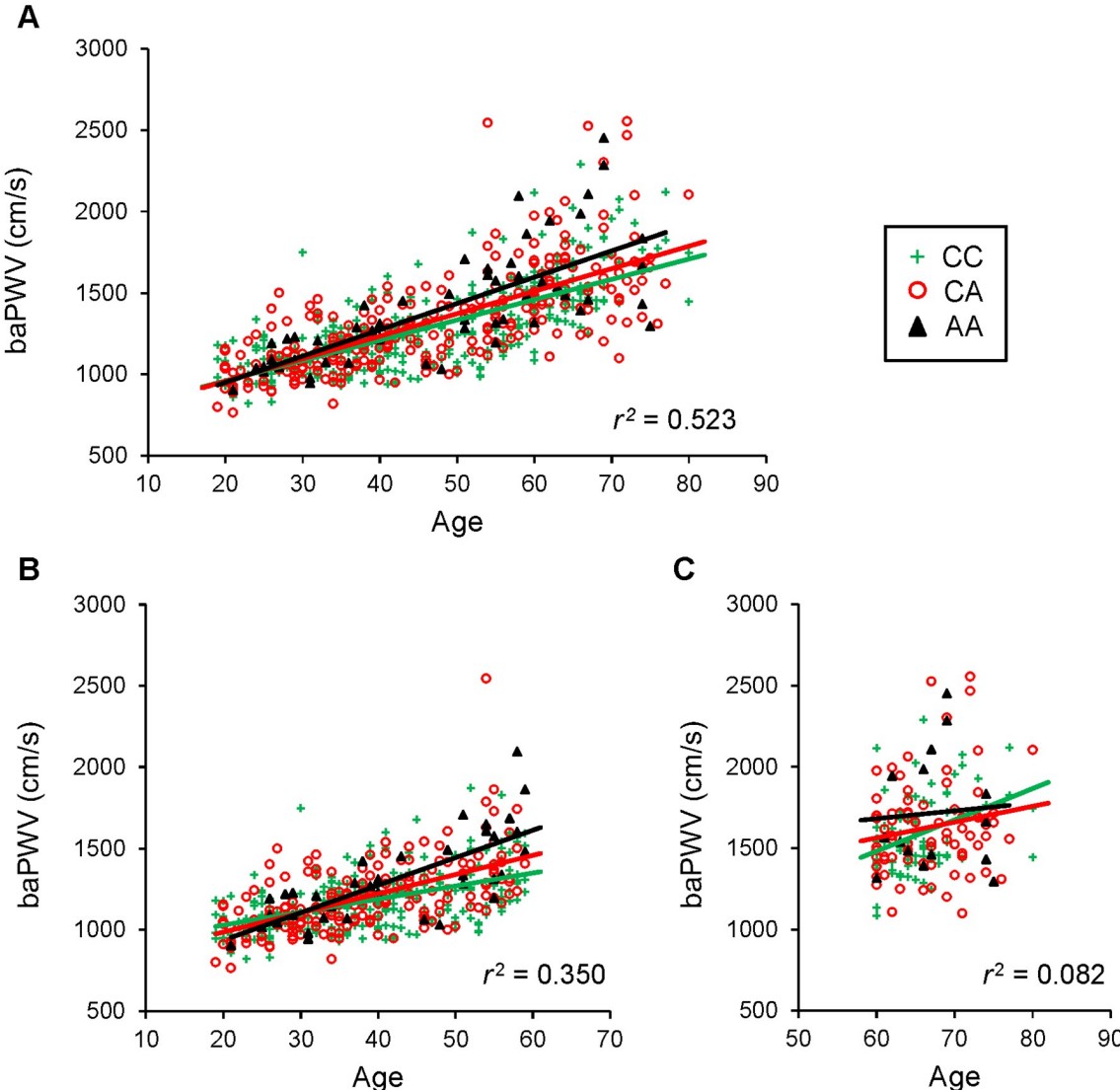

**Fig 3. Arterial stiffness and blood pressure were affected by the interaction between *NRF2* SNP−617 and age.** The scatterplot representing the relationship between baPWV and age divided into *NRF2* SNP−617 carriers (color: green, CC; red CA; black, AA) A) in all never-smoking healthy subjects, B) in those under 60 years old and C) in those over 60 years old. The coefficient of determination ($r^2$) was calculated by standardized multiple linear regression analyses for baPWV using 5 characteristics, i.e., *NRF2* SNP−617 (CC vs CA), SNP−617 (CC vs AA), age, interaction between SNP−617 (CC vs CA) and age, and interaction between SNP-617 (CC vs AA) and age.

### Association between *NRF2* promotor SNP−617 and arterial stiffness

This study demonstrated for the first time that baPWV was significantly higher in AA carriers than in CC carriers (Fig 2A). Interestingly, the ABI, which is an indicator of vascular lumen diameter, was not affected by SNP−617 (Fig 2B). Therefore, it was suggested that the association between SNP−617 and the increase in arterial stiffness is not due to atherosclerosis involving narrowing of the blood vessel lumen. Notably, although Nrf2 is an important protective factor against endothelial dysfunction, many studies including ours have reported that Nrf2 promotes atherosclerotic plaque formation in mice [49, 50]. Studies in mice showed that the

protective function of Nrf2 against early phase atherosclerosis formation via inhibition of inflammatory cell recruitment was canceled by the later accelerating function of Nrf2 in atherosclerotic plaque formation. Additionally, both MAP and PP, which are indicators of peripheral vascular resistance and stiffness of the elastic artery, respectively, were significantly higher in the order of CC, CA and AA (Fig 2E and 2F and S2 Table). These results suggest that SNP−617 affects peripheral small vessels as well as large vessels.

SBP, but not DBP, was significantly affected by SNP−617 (Fig 2C and 2D). As it is known that SBP is tightly correlated with PWV values both as a causative and a resultant factor, it is difficult to pinpoint the direct effect conferred by SNP−617. Actually, it is reported that in Japanese patients on hemodialysis, the SBP and DBP of female SNP−617 AA carriers were higher than those of CC + CA carriers [35], while the study of Japanese subjects who underwent a health evaluation at a hospital showed that the SBP and DBP of male SNP−617 CA+AA carriers were lower than those of CC carriers, but the DBP of female SNP−617 AA carriers was higher than that of CC+CA carriers [48]. These conflicts between the results of two studies were thought to be due to the incorporation of CA into CC or AA carriers. Our results showed that SNP−617 affects BP in healthy Japanese people and that the BP of not only AA but also CA increased compared with CC (Fig 2C and 2D and S2 Table). Our results suggest that CA functions as an intermediate type. As the CAVI (cardio ankle vascular index) can discriminate BP and arterial stiffness, the cause-effect relationship between SBP and baPWV should be further explored in the future. As accumulating reports have demonstrated the protective role of Nrf2 against dysfunction of endothelial cells, we surmise that SNP−617 affects arterial stiffness due to its direct effect on the vasculature itself and that hypertension is secondary to arterial stiffness.

## Association between SNP−617 and other physiological biomarkers

Consistent with our contention that SNP−617 directly affects arterial stiffness, biomarkers of general condition that may affect BP, such as erythropoiesis (erythrocyte, hemoglobin and hematocrit), liver function (total protein, AST, ALT and ©-GTP), renal function (BUN), insulin resistance (fasting glucose, HbA1c, insulin, HOMA-IR and C-peptide) and lipid metabolism (triglyceride, total cholesterol, LDL cholesterol and LH ratio), except for creatinine and HDL cholesterol, are not largely affected by SNP−617 (Table 2). Creatinine levels in SNP−617 AA carriers were significantly lower than those of CC or CA carriers. An increased serum creatinine level is an indicator of severe renal impairment and it usually does not elevate until renal function decrease by 50% [51]. As the medians of creatinine levels in each SNP−617 carriers in this study was within normal reference range (0.65–1.07 mg/dL for male and 0.46–0.79 mg/dL for female) [52], we surmise that the low creatinine level in AA is not related to the renal function. Consistent with this contention, BUN levels were not significantly different between subjects with different genotypes. Because creatinine is a product of muscle catabolism [53], AA carrier may have less muscle mass compared to other genotype carriers. This possibility remains to be clarified in the future studies. On the other hand, lipid metabolism is an important factor in considering the risk of CVD. To assess lipid metabolism, one important factor is the balance of LDL cholesterol and HDL cholesterol called LH ratio [54, 55]. If lipid metabolism had been affected by SNP−617, LH ratio should have provided clearer results than that of HDL cholesterol. However, the results of LH ratios were statistically less significant by the K.W. test (p = 0.092). Furthermore, there were no significant differences between the HDL cholesterol levels of each carrier by the multiple Mann-Whitney U test with Holm's method. On the other hand, the HDL cholesterol tends to be lower only in heterozygotes of SNP−617 compared to subjects of other genotypes and the same pattern of the difference was

recapitulated only in females by sex stratified analysis. Taken together, the difference of HDL cholesterol by SNP−617 genotypes, if any, should be vigorously confirmed in the future analysis.

## Differences in the effect of SNP−617 by sex

In the sex-stratified analysis, the female results were similar to the overall results (Table 3, S2 and S3 Tables). However, there were no significant differences in the male results (S2 and S4 Tables). We surmise that there are two factors that hamper the proper comparison of the male with female subjects: selection bias by different exclusion rates and insufficient statistical power. The exclusion rates for male were 80.2% (560 excluded subjects/698 base subjects) and 54.3% (598 excluded subjects/1102 base subjects) for female, and there was significant difference between them ($\chi$2 test's p value < 0.001). In Japan, employers are required by law to provide regular health checkups for the employees, and the employment rate of males is generally higher than that of women. Therefore, since males are more likely than females to have a health checkup, it is easier for males to notice their own diseases, and as a result, it is expected that more males met the exclusion criteria. In addition, more males were excluded from the analysis because of the significantly higher rate of smoking history in males ($\chi$2 test's p value < 0.001. Number of subjects with smoking history/base subjects: males, 452/698; females, 238/1102). Because of the high exclusion rate, only 10 AA male carriers remained in the analysis. This is an apparent limitation of the research and further research with additional subjects is needed to clarify this point.

## Effect of SNP−617 on age-dependent increase of baPWV

Multivariate analysis revealed that the age-dependent incremental ratio in baPWV was higher in the order of AA, CA and CC (Fig 3A and Table 3). Three mechanisms lead to cellular senescence: telomere-dependent replicative senescence, oncogene-induced senescence and stress-induced premature senescence [24]. As SNP−617 affects baPWV in individuals under 60 years of age, it is an interesting hypothesis that low NRF2 activity induces premature vascular senescence.

The relationship between SNP−617 and baPWV was more pronounced in people under 60 years old, but there was no significant difference in the relationship in those over 60 years old (Fig 3B and 3C and Table 3). We consider that the selection bias and the insufficient statistical power discussed above also affected the analysis of subjects over 60 years old. For those under 60 years of age, 56.7% were excluded from the analysis, while 75.8% of those over 60 years of age were excluded from the analysis by our criteria (Number of excluded/base subjects: < 60 years old, 611/1078; $\geq$ 60 years old, 547/722, respectively), and there was significant difference between them ($\chi$2 test's p value < 0.001). This raises concerns that those with advanced arteriosclerosis were excluded from the analysis. Also, SNP−617 AA carriers over age 60 were small in number (i.e., only 15 AA carriers), which precluded more detailed age stratification analysis. These are the apparent limitations of this study and further studies are required to elucidate the relationship of SNP−617 with people over 60 years of age.

## Conclusions

This study provided important evidence for the association of NRF2 activity and arterial stiffness. We have shown that not only AA but also CA contributes to arterial stiffness and BP in never-smoking healthy subjects. Furthermore, it was suggested that the effect of SNP−617 on arteriosclerosis was not accompanied by narrowing of the blood vessel lumen, which is generated mainly via atherosclerosis. In addition, we revealed that the interaction of SNP−617 and

aging affects increasing arterial stiffness by multivariate analysis. However, this study did not reveal this relationship in those over 60 years of age. In the future, it is expected that the prevention and treatment of arteriosclerosis by targeting NRF2 will be developed based on the results of this study.

## Supporting information

**S1 File. The questionnaires of self-reported sex, age, clinical history, medication history of hypertensive medicine and smoking history in Iwaki Health Promotion Project.**
(PDF)

**S1 Table. The frequency of the NRF2 SNP (−617C>A) stratified by gender.** [a] *P* values were calculated by χ2 test.
(PDF)

**S2 Table. The median and IQR of four-limb blood pressure measurements in never-smoking healthy subjects.** [a] *P* values were calculated by Kruskal-Wallis (K.W.) test with post-hoc test by Holm. Abbreviation: IQR, interquartile range; baPWV: bronchial-ankle pulse wave velocity, ABI: ankle branchial pressure index, SBP: systolic blood pressure, DBP: diastolic blood pressure, MAP: mean arterial pressure, PP: pulse pressure.
(PDF)

**S3 Table. The median and IQR of general characteristics in female never-smoking healthy subjects.** [a] *P* values were calculated by Kruskal-Wallis (K.W.) test with post-hoc test by Holm. Abbreviations: IQR, interquartile range; AST, aspartate transaminase; ALT, alanine aminotransferase; γ-GTP, gamma-glutamyl transpeptidase; BUN, blood urea nitrogen; HbA1c, glycated hemoglobin; HOMA-IR, homeostasis model assessment of insulin resistance; HDL, high-density lipoprotein-cholesterol; LDL, low-density lipoprotein-cholesterol; L/H ratio, LDL/HDL ratio.
(PDF)

**S4 Table. The median and IQR of general characteristics in male never-smoking healthy subjects.** [a] *P* values were calculated by Kruskal-Wallis (K.W.) test with post-hoc test by Holm. Abbreviations: IQR, interquartile range; AST, aspartate transaminase; ALT, alanine aminotransferase; γ-GTP, gamma-glutamyl transpeptidase; BUN, blood urea nitrogen; HbA1c, glycated hemoglobin; HOMA-IR, homeostasis model assessment of insulin resistance; HDL, high-density lipoprotein-cholesterol; LDL, low-density lipoprotein-cholesterol; L/H ratio, LDL/HDL ratio.
(PDF)

## Acknowledgments

The authors thank all the members of Hirosaki COI for their skillful contributions to the collection and management of the data.

## Author Contributions

**Conceptualization:** Sunao Shimizu, Shigeyuki Nakaji, Ken Itoh.

**Data curation:** Sunao Shimizu, Ken Itoh.

**Formal analysis:** Sunao Shimizu, Takanori Hasegawa, Eigo Shimizu, Seiya Imoto, Masayuki Yamamoto, Ken Itoh.

**Funding acquisition:** Sunao Shimizu, Shigeyuki Nakaji, Ken Itoh.

**Investigation:** Sunao Shimizu, Junsei Mimura, Shuya Kasai, Ken Itoh.

**Methodology:** Sunao Shimizu, Seiya Imoto, Ken Itoh.

**Project administration:** Sunao Shimizu, Seiya Imoto, Hiromi Yamazaki, Yusuke Ushida, Ken Itoh.

**Supervision:** Sunao Shimizu, Michiko Tsushima, Hiroyuki Suganuma, Shigeyuki Nakaji, Ken Itoh.

**Validation:** Sunao Shimizu, Ken Itoh.

**Visualization:** Sunao Shimizu, Ken Itoh.

**Writing – original draft:** Sunao Shimizu, Ken Itoh.

**Writing – review & editing:** Sunao Shimizu, Michiko Tsushima, Shuya Kasai, Hiromi Yamazaki, Hirofumi Tomita, Ken Itoh.

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
