## [Decision Letter · Decision Letter 0]

27 May 2020

PONE-D-20-12583

Association of single nucleotide polymorphisms in the NRF2 promoter with vascular stiffness with aging

PLOS ONE

Dear Dr. Itoh,

Thank you for submitting your manuscript to PLOS ONE. After careful consideration, we feel that it has merit but does not fully meet PLOS ONE’s publication criteria as it currently stands. Therefore, we invite you to submit a revised version of the manuscript that addresses the points raised during the review process.

ACADEMIC EDITOR:

Thank you for submitting your manuscript to PLoS ONE. Although the evaluations by two reviewers were favorable, there are several comments and questions. I would like you to check the manucript carefully and respond to all of them.

We look forward to receiving your revised manuscript.

Kind regards,

Hiroyuki Itabe, Ph.D.

Academic Editor

PLOS ONE

Journal Requirements:

4. Thank you for stating the following in the Financial Disclosure section:

"This work was supported by "Center for Innovation Program" of the Japan Science and Technology Agency Grant Number JPMJCE1302 (to SN and KI).

SN: Shigeyuki Nakaji, KI: Ken Itoh.

URL (https://www.jst.go.jp/EN/index.html)

We note that one or more of the authors are employed by a commercial company: 'Kagome Co., Ltd.'

Reviewers' comments:

Reviewer's Responses to Questions

**Comments to the Author**

1. Is the manuscript technically sound, and do the data support the conclusions?

Reviewer #1: Yes

Reviewer #2: Yes

2. Has the statistical analysis been performed appropriately and rigorously? 

Reviewer #1: Yes

Reviewer #2: Yes

3. Have the authors made all data underlying the findings in their manuscript fully available?

Reviewer #1: Yes

Reviewer #2: Yes

4. Is the manuscript presented in an intelligible fashion and written in standard English?

Reviewer #1: Yes

Reviewer #2: Yes

5. Review Comments to the Author

Reviewer #1: Authors conducted well designed study with enough subject number. Results from their study enhanced new insight into roll of Nrf2 in atherosclerosis. This reaches standards of published in peer-reviewed journal. Thank you.

Reviewer #2: The authors examined baPWV, an indicator of vascular stiffness and SNPs in the NRF gene SNP-617 and showed the clear association of NRF activity and arterial stiffness. However, some issues written below are still concerned.

Major:

1. The authors showed the clear relationship between baPWA and SNP-617 in people under 60 years old. However, if it is true, this relationship should become clearer in people over 60 years old, which may become more important in terms of prevention of cardiovascular disease.

2. Although the authors described in the limitation of the study, it is not clear whether there is a gender difference in the results of baPWV, SBP, MAP and PP.

3. It is relatively easy to understand there is a relationship between SNP-617 and arterial stiffness, but hard to connect it to creatinine and HDL cholesterol levels.

Minor:

1. P6, l11 oxidative stress: under the condition of oxidative stress?

2. P10, l5 SBP pressure: delete pressure

3. P13, l14 CA, 1317/1169-1589: change to AA,

4. P14, l16 *p<0.01: change to **p<0.01

5. P22, l16 creatinine HDL cholesterol: insert and between creatinine and HDL cholesterol

6. P22, l18 AA, CA, and CC: change to AA, CA and CC

6. PLOS authors have the option to publish the peer review history of their article (what does this mean?). If published, this will include your full peer review and any attached files.

Reviewer #1: No

Reviewer #2: No

---

## [Author Response · Author response to Decision Letter 0]

16 Jun 2020

Dear Dr. Itabe,

Thank you for your email of 27th May 2020 regarding our manuscript entitled "Association of single nucleotide polymorphisms in the NRF2 promoter with vascular stiffness with aging " and for providing us an opportunity to revise it. 

We appreciate for the insightful and straightforward comments from the reviewers. Responding to the reviewers’ comments, we added the description to fully discuss the reviewers’ concerns in the “Discussion” section. As a result, we believed that the manuscript has significantly improved. Please refer to the following “response to the reviewers’ comments” for the details.

I appreciate for your kind consideration of the revised manuscript and look forward to hearing from you at your earliest convenience.

Yours sincerely

Response to reviewers’ comments

Reviewer #1

Authors conducted well designed study with enough subject number. Results from their study enhanced new insight into roll of Nrf2 in atherosclerosis. This reaches standards of published in peer-reviewed journal. Thank you.

Response: Thank you for the comment and we appreciate for your evaluation.

Reviewer #2

The authors examined baPWV, an indicator of vascular stiffness and SNPs in the NRF gene SNP-617 and showed the clear association of NRF activity and arterial stiffness. However, some issues written below are still concerned.

Response: Thank for your evaluation of the manuscript and insightful and straightforward comments.

Major:

1. The authors showed the clear relationship between baPWA and SNP-617 in people under 60 years old. However, if it is true, this relationship should become clearer in people over 60 years old, which may become more important in terms of prevention of cardiovascular disease.

Response: Thank you for the comment. We consider that there are two main reasons why the relationship between baPWV, SNP−617 and age are not significant over 60 years old. One is selection bias and the other is insufficient statistical power. Accordingly, we added the description in the Discussion section (page 25, sentence 430 to 440 in the revised manuscript with track change).

2. Although the authors described in the limitation of the study, it is not clear whether there is a gender difference in the results of baPWV, SBP, MAP and PP.

Response: Thank you for the comment. Because the results in male contain similar analytical limitations as the analysis of the older subjects (i. e. selection bias and insufficient statistical power), we consider that this study couldn’t reveal gender differences in relationship between SNP−617 and baPWV. Therefore, we added the explanation in Discussion section (page 24, sentence 407 to 420 in the revised manuscript with track change)

3. It is relatively easy to understand there is a relationship between SNP-617 and arterial stiffness, but hard to connect it to creatinine and HDL cholesterol levels.

Response: Thank you for the comment. We have added some additional discussion about the results of creatinine and HDL cholesterol in the Discussion section (page 23, sentence 383 to 403 in the revised manuscript with track change).

Minor:

1. P6, l11 oxidative stress: under the condition of oxidative stress?

Response: Thank you for the comment. We have corrected the manuscript.

2. P10, l5 SBP pressure: delete pressure

Response: Thank you for the comment. We have corrected the manuscript.

3. P13, l14 CA, 1317/1169-1589: change to AA,

Response: Thank you for the comment. We have corrected the manuscript.

4. P14, l16 *p<0.01: change to **p<0.01

Response: Thank you for the comment. We have corrected the manuscript.

5. P22, l16 creatinine HDL cholesterol: insert and between creatinine and HDL cholesterol

Response: Thank you for the comment. We have corrected the manuscript.

6. P22, l18 AA, CA, and CC: change to AA, CA and CC

Response: Thank you for the comment. We have corrected the manuscript.

---

## [Decision Letter · Decision Letter 1]

15 Jul 2020

Association of single nucleotide polymorphisms in the NRF2 promoter with vascular stiffness with aging

PONE-D-20-12583R1

Dear Dr. Itoh,

We’re pleased to inform you that your manuscript has been judged scientifically suitable for publication and will be formally accepted for publication once it meets all outstanding technical requirements.

Kind regards,

Hiroyuki Itabe, Ph.D.

Academic Editor

PLOS ONE

Reviewers' comments:

Reviewer's Responses to Questions

**Comments to the Author**

1. If the authors have adequately addressed your comments raised in a previous round of review and you feel that this manuscript is now acceptable for publication, you may indicate that here to bypass the “Comments to the Author” section, enter your conflict of interest statement in the “Confidential to Editor” section, and submit your "Accept" recommendation.

Reviewer #2: All comments have been addressed

2. Is the manuscript technically sound, and do the data support the conclusions?

Reviewer #2: Yes

3. Has the statistical analysis been performed appropriately and rigorously? 

Reviewer #2: Yes

4. Have the authors made all data underlying the findings in their manuscript fully available?

Reviewer #2: Yes

5. Is the manuscript presented in an intelligible fashion and written in standard English?

Reviewer #2: Yes

6. Review Comments to the Author

Reviewer #2: The authors have appropriately addressed all comments the reviewers made and have revised the manuscript well.

7. PLOS authors have the option to publish the peer review history of their article (what does this mean?). If published, this will include your full peer review and any attached files.

Reviewer #2: No

---

## [Editor Report · Acceptance letter]

30 Jul 2020

PONE-D-20-12583R1 

Association of single nucleotide polymorphisms in the NRF2 promoter with vascular stiffness with aging 

Dear Dr. Itoh:

I'm pleased to inform you that your manuscript has been deemed suitable for publication in PLOS ONE. Congratulations! Your manuscript is now with our production department. 

Kind regards, 

on behalf of

Dr. Hiroyuki Itabe 

Academic Editor

PLOS ONE